# Perception of Peruvian Students Studying in Biological Sciences about the Advantages of Virtual Classes during the COVID-19 Pandemic

Aldo Bazán-Ramírez [1,*], Walter Capa-Luque [2], Homero Ango-Aguilar [3], Roberta Anaya-González [3] and Víctor Cárdenas-López [3]

1 Escuela de Medicina, Universidad César Vallejo, Trujillo 13007, Peru
2 Facultad de Psicología, Universidad Nacional Federico Villarreal, Lima 15082, Peru; wcapa@unfv.edu.pe
3 Facultad de Ciencias Biológicas, Universidad Nacional de San Cristóbal de Huamanga, Ayacucho 05001, Peru; homero.ango@unsch.edu.pe (H.A.-A.); roberta.anaya@unsch.edu.pe (R.A.-G.); victor.cardenas@unsch.edu.pe (V.C.-L.)
* Correspondence: abazanramirez@gmail.com

**Abstract:** There is significant educational research interest regarding the assessment of the benefits of virtual education implemented during the COVID-19 pandemic for university programs that were essentially face-to-face. The objective of the study was to determine the effect of the management of online resources that the teacher had on the valuation of the advantages of online classes in biological sciences, mediated by the students' perception of virtual practices as well as the accessibility and use of online resources. A total of 332 Peruvian students studying in biological sciences from a public university, enrolled in five undergraduate academic years, of which 184 were women and 148 were men, participated. A non-experimental predictive design of causal relationships was used with the methodology of structural equation modeling. According to the SEM model (CFI and TLI > 0.95, RMSEA and SRMR < 0.05), the valuation of the advantages of virtual classes during the pandemic was significantly predicted by the valuation of virtual practices (positively) as well as by the accessibility and management of online resources by students (negatively); likewise, the use and mastery of digital and online resources by teachers had an indirect effect on the valuation of virtual classes, but direct effects on virtual practices and accessibility to digital resources by students. Also, virtual practice was the most crucial variable in predicting the valuation of online classes ($\beta = 0.48$, $p < 0.001$). In conclusion, the student's perception of the teachers' handling of online resources during the COVID-19 pandemic was determinant as a favorable valuation of the advantages offered by online classes, a relationship that is mediated by virtual practices and accessibility to online resources.

**Keywords:** COVID-19 pandemic; biological science students; online classes; teacher



## 1. Introduction

The crisis caused by the COVID-19 pandemic has deeply affected the education system, and against this adversity, educational modalities that were essentially classroom-based were transformed into virtual classes worldwide, especially at the undergraduate level in universities. In this context of confinement and social asylum, the use of e-learning, digital learning, and digital communication allowed higher education institutions to contribute to the development of educational sustainability.

E-learning was established as a new way of implementing the learning process in the university context [1,2]. Thus, the complete transition to virtual classes was considered a great alternative and a means to promote better student participation [3]. A study in Greece in the first two months of the COVID-19 pandemic confirmed great enthusiasm among university students for these new ways of virtual teaching and learning [4]. This initial optimism led to the assumption that the pandemic enabled the integral digital

transformation of universities, favorably affecting teaching and learning at the university level [5,6].

In this new virtual teaching modality, e-learning studies and innovations have been reported in various parts of the world to improve the quality of the education system and achieve student satisfaction [7,8], facilitated in part by collaborative learning [9] and self-regulated learning [10], mediated by computer, and information and communication technologies [11]. It also evidenced greater effectiveness of synchronous learning for small-group, case-based, highly interactive sessions that teach synthesis and application [12].

University students' perception of the usefulness and advantage of virtual classes during the COVID-19 pandemic is an essential point in educational researchers' reflections worldwide. For example, a study with undergraduate students from India during the first semester of 2020 showed a good acceptance of e-learning in times of crisis [2]. Another study in India showed that university students valued virtual classes for their functionality during the confinement period, even though the classes themselves were affected by network problems, data card problems, and power outage issue [13].

United Arab Emirates students valued virtual classes as a medium that interestingly provided scientific material and increased the possibility of contact between students and between students and teacher during the social isolation due to the pandemic [14]. In another study with university students in Indonesia during the pandemic, it was found that most students considered virtual classes to be very helpful [15]. Likewise, Maatuk et al. [16] reported that students and teachers of information technology at a university in Libya valued, in a more significant proportion, the advantages of e-learning in virtual classes due to the pandemic. Also, Saudi Arabian university students considered virtual teaching a better experience for the students who commute, greater involvement with students, better interaction, more effective use of technology, and improved learning results [17].

This assessment of university students' evaluations of the usefulness and advantages of virtual education in times of the pandemic has received significant attention from educational researchers worldwide. The interest goes beyond some models of adopting new technologies at the higher education level to explain how students value the advantages of virtual education favorably in educational programs that were essentially face-to-face before the pandemic. Among these explanatory models of the use of new technologies in the educational field, applications have been made, for example, the self-determination theory [18] or the technology adoption model (TAM) [18–22]. However, the explanation of the students' judgment on the advantages and usefulness of virtual classes has been raised, without aligning themselves theoretically or methodologically to the cognitive models of technology adoption in virtual learning, especially in times of pandemic.

What can be seen so far is that various factors have been included to describe student perceptions of virtual instruction [14–22]. A central aspect has been to consider the satisfaction, usefulness, and effectiveness of learning in virtual classes for students. Another aspect to highlight is the inclusion of associated variables or predictors of these aspects to factors of accessibility and the availability of technological and online devices and resources by students; psychological aspects of students, e.g., motivation, disposition, involvement, self-efficacy, and social interaction; aspects of management of technological and online resources by students, for example, adaptation to virtual channels and media, digital skills, knowledge of technological alternatives as complementary for their learning; and the digital and didactic competence of teachers during virtual classes in the pandemic.

For the present study, we take up three variables that can explain university students' perception of the usefulness and advantages of virtual classes in a pandemic: management of online resources by the teacher, virtual practices (teaching), and student accessibility (and digital skills). Below, we will describe some of these variables that may be associated with students' perceptions of the advantages of virtual classes.

## 1.1. The Professor's Performance and Virtual Resources Domain

Student approval of the usefulness and advantages of virtual classes during the COVID-19 pandemic has been associated with teacher competencies in virtual education and mastery of resources for online teaching. In this way, the effect of the instructional practice, the professor's performance, domain and management of digital resources on student satisfaction and positive assessment regarding the use of these virtual courses and on their learning has been documented.

Moorhouse et al. [23] have pointed out that, in virtual teaching, teachers must develop at least three e-classroom interactional competencies: technological, online environment management, and online interaction competencies. However, these teacher skills can differentially influence students' perceptions regarding the usefulness and advantages of virtual learning. For example, with Chinese university students, Wang et al. [24] reported a significant and negative effect of the professor's performance on satisfaction with online learning and on learning outcomes, and a positive and significant effect of the professor's innovation on satisfaction with online learning and the learning outcomes. Likewise, in Peruvian postgraduate students, virtual teaching practices had significant effects on the participation and learning of the students; in turn, it significantly affected the results of applying what was learned [25].

With undergraduate and graduate students from Malaysia, Mohammed et al. [8] found that teacher performance, course evaluation and students' factors had significant effects on student satisfaction about virtual learning. Student factors were a latent second-order factor, but the authors erroneously included a first-order factor, Student–Instructor Interaction, since it describes instructional relationships and is related with teacher–student didactic performance [25]. Perhaps a latent second-grade factor, including three instructional constructs (teacher performance, course evaluation, teacher–student interaction), would have a greater capacity to explain student satisfaction.

Other aspects that influence the student's perception and usefulness of virtual classes are the quality of instruction (quality of information), teaching capacity, and instructional support. Rokhman et al. [26] found that with university students in Indonesia, the quality of the information provided in the virtual mode and the capacity of teachers significantly predicted satisfaction with the use of virtual learning. On the other hand, with university students from Taiwan, Zhao et al. [27] reported that support in the management of virtual return, cognitive support, and emotional support are significant predictors of virtual learning satisfaction. With students from Pakistan, Younas et al. [28] reported the significant effect of the adequacy of online education channels on e-learning satisfaction during the COVID-19 pandemic.

With Romanian university students who utilized e-learning during the pandemic, Fülöp et al. [29] found that variables such as course content and design, and instructor contribution significantly predicted e-learning satisfaction to academic success, mediated by perceived utility (only in the case of course content and design) and by perceived ease of use, in the case of the three variables (course content and design, instructor contribution, and quality of the e-learning system).

## 1.2. Accessibility to the Virtual Environment and Digital Competencies of the Student

The students' perception of the advantages, effectiveness, and usefulness of virtual classes in pandemics have been predicted significantly by the perception of accessibility to the virtual environment and competencies in the domain of information technologies on the part of students [30,31]. The study by Bazán et al. [30] was conducted with Peruvian graduate students in educational sciences, while the study reported by Pomares et al. [31] was conducted with Cuban graduate students in medical sciences in the first year of the COVID-19 pandemic.

Libyan students, who took virtual classes during the pandemic, reported difficulty in accessing e-learning, and the low quality of internet services as primary obstacles to the application of virtual learning [16]. Likewise, a study with Malaysian university students

found that the quality of the virtual system directly and significantly predicted student satisfaction regarding online learning [8]. The importance of the quality of the e-learning system as a significant predictor of online learning satisfaction was also confirmed in Indonesian students [26].

As with Peruvian and Cuban students, Indonesian students found that student ability (digital skills) was associated with students' satisfaction with virtual classes [26,30,31]. Likewise, in a study with university students in Pakistan, significant effects were found on e-learning satisfaction, as well as factors related to the domain from online and digital resources (digital competence) [28].

With regards to Romanian university students taking classes online during the pandemic, it was found that the capacity for the use of digital resources, the quality of the e-learning system, and previous experience in e-learning significantly predicted perceived ease of use, and this, in turn, significantly predicted satisfaction and personal development by online classes [29]; that is, these three variables related to digital skills were influenced indirectly and significantly on previous experience in e-learning. In virtual classrooms of Chinese college students during the pandemic, satisfaction with virtual classes was significantly explained by the perceived usefulness of virtual classes [32]; however, the system quality variable did not have a significant effect on perceived utility.

### 1.3. The Practical Teaching and Laboratory Practices

One aspect to consider is the practical teaching in the virtual modality in disciplines that require laboratory practices when university facilities remained closed during the pandemic. As another researcher pointed out, the optimal method to support online disciplinary practices needs more publication [33]. Therefore, in research on the reception of virtual classes during the pandemic with students from disciplines that require practice, questions about that experience should be included during virtual practices. Virtual education during the pandemic led to the design and adaptation of a practical on-site laboratory in the online modality, accompanied by active methodologies for learning online practices [34].

The case of laboratory practices in teaching biological sciences also involved adapting face-to-face laboratory activities to the virtual modality. Biology labs took much work to handle as students needed help to conduct experiments physically in laboratory classrooms [35]. Among other findings, an experience of the implementation of a simulation and research laboratory of a higher degree course in genetics in the United States was reported, in which advantages of online teaching were observed, such as the detailed annotation of the explanations, the high quality of the class slides and the deepening of essential conceptual aspects [35].

A study with pre- and post-test biological science students in Indonesia, which included home practices to apply new and different adaptations to a face-to-face practice in the laboratory, in topics of respiration, photosynthesis, food nutrients, and contaminants fostered students' autonomy and creativity in collecting materials, modifying tools, preparing practice worksheets, and choosing practice sites [36]. In Canada, with undergraduate biology students from a semester course in plant physiology, a home lab was implemented to study the impact of nitrogen addition on growth rates and the root nodulation of wild nitrogen-fixing rhizobia on Pisum Sativum (pea) plants. This home lab allowed students to develop research in a hands-on way, with the flexibility to collect and analyze their data in a remote environment during the COVID-19 pandemic [37]. Likewise, with undergraduate and graduate students in Wisconsin (United States), innovation and adaptation to an online learning environment of molecular bioscience laboratories in the specialties of genetics, cell biology, bioinformatics, and advanced microscopy were reported (virtual teaching induced by the pandemic stimulated some innovations in the teaching of practices, which improved instruction in these areas of science) [33].

As in other types of virtual classes during the pandemic, laboratory practices were guided virtually, with video tutorials and live virtual sessions [37]. Difficulties were also

encountered despite their strength and effectiveness; virtual practices cannot replace practical experience in conducting actual biological experiments to master technical skills [35]. In addition, these virtual practices during the pandemic deepened the inequity of learning opportunities among students, as marginalized students (of low socioeconomic status) studying science suffered a disproportionate burden from COVID-19 and the shift to an online learning environment [33].

### 1.4. This Research

Based on the background presented, the research reported here was conducted to fill knowledge gaps to predict the valuation of the benefits of online classes in the context of a pandemic. To this end, the following research question was formulated: what is the effect of the teacher's management of online resources (as perceived by biological science students) on the valuation of the advantages of online classes, mediated by the self-perception of virtual practices as well as by the accessibility and use of online resources?

Likewise, the objective of this research was to determine the effect of the teacher's management of online resources on the valuation of the advantages of online classes in biological sciences, mediated by the students' perception of virtual practices as well as by the accessibility and use of online resources.

## 2. Materials and Methods

### 2.1. Research Design

Due to the nature of the causal relationships between the variables, the study is a multivariate, non-experimental, cross-sectional study.

### 2.2. Participants

Of all the students of the Professional School of Biology of the Faculty of Biological Sciences of the Universidad Nacional Mayor de San Cristóbal de Huamanga that registered, 552 students enrolled in the 2020-I semester in the 2nd, 4th, 6th, 8th and 10th cycles of the 2004 Readjusted Study Plan were invited to participate in the study by means of an e-mail. A total of 332 students, who accepted a virtual invitation and sent the duly completed and signed informed consent form, participated in this study. The inclusion criteria were as follows: to be formally enrolled, to have completed the 2020-I semester, to have at least 80% online attendance in the course and to have signed the informed consent form.

### 2.3. Ethical Considerations

The protocol of this investigation was approved by the Institutional Review Board of Sciences Biological Faculty of Universidad Nacional de San Cristobal de Huamanga. Furthermore, this investigation was conducted in accordance with the Declaration of Helsinki. The non-coercion of the participants was also guaranteed at the time of recruitment and/or at the time of signing the informed consent. The invitation was emailed, emphasizing the students' voluntary nature and selfless participation. Furthermore, they were told that there are no negative consequences should they decide not to participate in the study.

### 2.4. Study Context

The study was conducted in the region of Ayacucho, located in a socioeconomic context of extreme poverty, which is reflected in the lack of adequate connectivity to internet services. In addition to this, there is the need for more technological resources such as the latest model laptops, internet access, etc. In the Professional School of Biology of the Faculty of Biological Sciences of the National University of San Cristóbal de Huamanga, many students are in their places of origin or work centers, far from the university's headquarters. On the other hand, most students come from lower socioeconomic levels, so their economic income does not allow them to implement computer resources and manage technological resources, hindering the quality of their academic training.

### 2.5. Measuring Instrument and Materials

Online Class Assessment Questionnaire: A questionnaire was constructed in a Likert scale format, designed based on the literature available as of August 2020, on some aspects of online learning in the context of confinement by SARS-CoV-2/COVID-19, mainly taking as a reference to the studies of Bazán-Ramírez et al. [30], Owusu-Fordjour et al. [38], Pomares-Bory et al. [31], and Cheng [39]. The student's answers to each question (statements) were on a scale ranging from 0 to 3; 0 = Never, 1 = Sometimes, 2 = Almost always, and 3 = Always.

This questionnaire was subjected to construct a validity analysis considering four dimensions (constructs): 1. Use of online resources by the teacher. 2. Accessibility and resources, statements. 3. Virtual practice. 4. Evaluation of Online classes. Figure 1 shows the confirmatory factor analysis performed to obtain convergent and divergent construct validity. Appendix A presents the questionnaire organized in the four dimensions indicated with 16 items. Figure 1 presents the confirmatory factor analysis model obtained for convergent and divergent validation of the constructs in measuring the perception of virtual classes in the biological sciences. The confirmatory model evaluated with the software R using WLSMV as an estimator given the ordinal category of the items presented the following appropriate goodness of fit indices: $\chi^2$ (98) = 198.743, $p < 0.001$, CFI = 0.972, TLI = 0.966, RMSEA = 0.056 [0.045, 0.067], SRMR = 0.053 y WRMR = 0.897. The analysis of the parameters related to the standardized factor loads also denoted good relationships between the items and the corresponding factors since the loads varied between 0.50 and 0.88. These data support the existence of validity based on the internal structure for questionnaire scores.

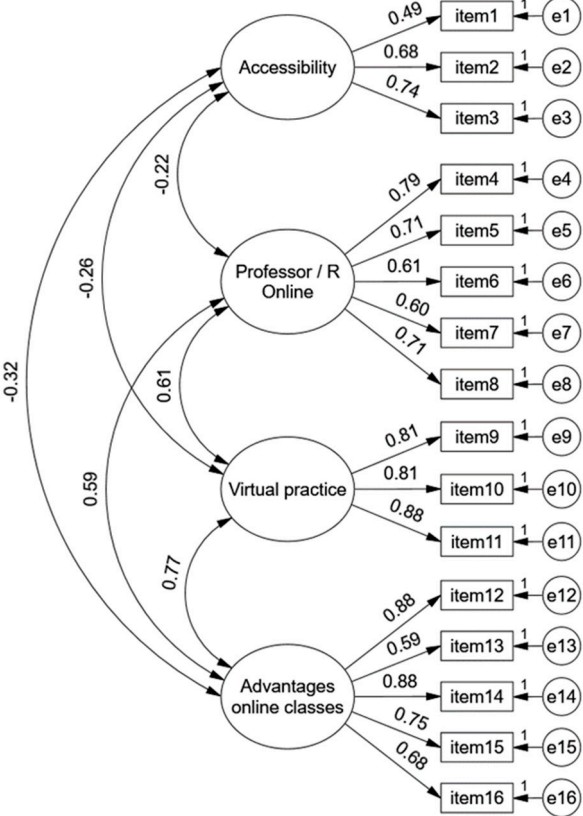

**Figure 1.** Factorial structure of the questionnaire of conditions and evaluation of online classes.

The reliability for the factor "use of online resources by the teacher" showed an omega coefficient of 0.85 and an ordinal alpha coefficient of 0.82; the "accessibility and resources factor" presented an omega coefficient of 0.67 and an ordinal alpha coefficient of 0.67; the virtual practice factor reached an omega coefficient of 0.87 and an ordinal alpha coefficient

of 0.86, and the "rating factor of online classes" presented an omega coefficient of 0.88 and an ordinal alpha coefficient of 0.87. In its complete version, the questionnaire presents a McDonald's omega coefficient of 0.92 and an ordinal alpha coefficient of 0.88, denoting these data as highly reliable for the instrument's measurements.

### 2.6. The Model Hypothetic

Based on the analyses and approaches presented in the introduction, a model of structural relationships is established as a general hypothesis (Figure 2) that postulates the following: If the teacher is competent and efficiently manages on-line resources, then the valuation attributed by students to the advantages of online classes is positive; this positive valuation is affected at the same time by the mediation of favorable virtual practices as lower difficulties for the accessibility and use of self-perceived online resources.

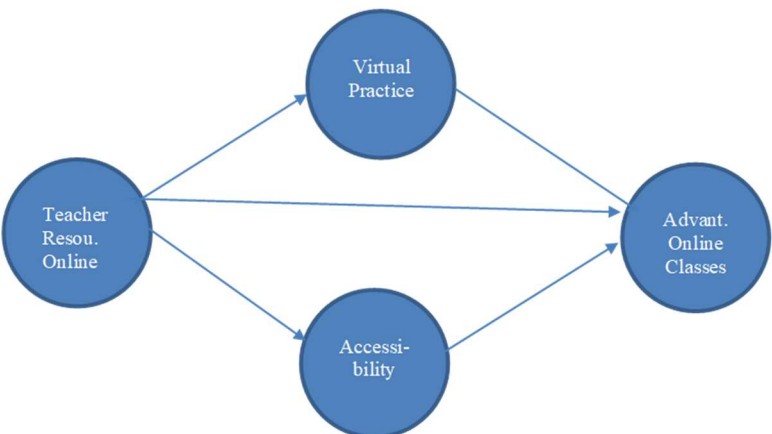

**Figure 2.** Hypothetical model.

### 3. Results

Figure 3 shows the model obtained from structural relationships between the teacher's use of online resources and the assessment of advantages of online classes, perceived by students and mediated by the constructs of accessibility and online practices. Using the R program and MLM estimator, a model with good goodness of fit was found (SB-$\chi$2 (97) = 153.649, $p$ = 0.000, CFI = 0.963, TLI = 0.954, RMSEA = 0.042 [0.029, 0.054], SRMR = 0.043). This resulting model shows that virtual practices significantly explain the advantage assessment factor of online classes (regression coefficient = 0.60, $p < 0.001$) and accessibility (negatively and with regression coefficient = 0.14, $p < 0.05$).

On the other hand, although the factor management of online resources by the teacher had a significant and positive effect on virtual practices ($\beta$ = 0.63, $p < 0.001$) and a significant and negative effect on accessibility ($\beta$ = $-0.20$, $p < 0.05$), its direct effect on the latent variable Advantages of Online Classes was significant. Between the use of online resources by the teacher and the advantages of online classes, there was a direct effect (positive and with regression coefficient = 0.19, $p < 0.05$).

A linear regression model with all predictor variables was also tested to identify an essential factor to explain the valuation of the advantages offered by online classes.

The recommended basic assumptions were examined for a correct interpretation of the multiple linear regression analysis [40]. The Durbin–Watson test of 2.005 was within the expected thresholds, ruling out the presence of autocorrelation. Multicollinearity was also not found because the tolerance values fluctuated between 0.76 and 0.97 (higher than the critical value of 0.10) while the VIF values (variance inflation factor) were between 1.03 and 1.30, which in no case exceeded the critical value of 10. Finally, the analysis of the means of standardized and non-standardized residues equal to zero denoted compliance with the assumption of multivariate normality, which was also confirmed with the Shapiro–Wilk multivariate test (S-W = 0.996, $p$ = 0.650).

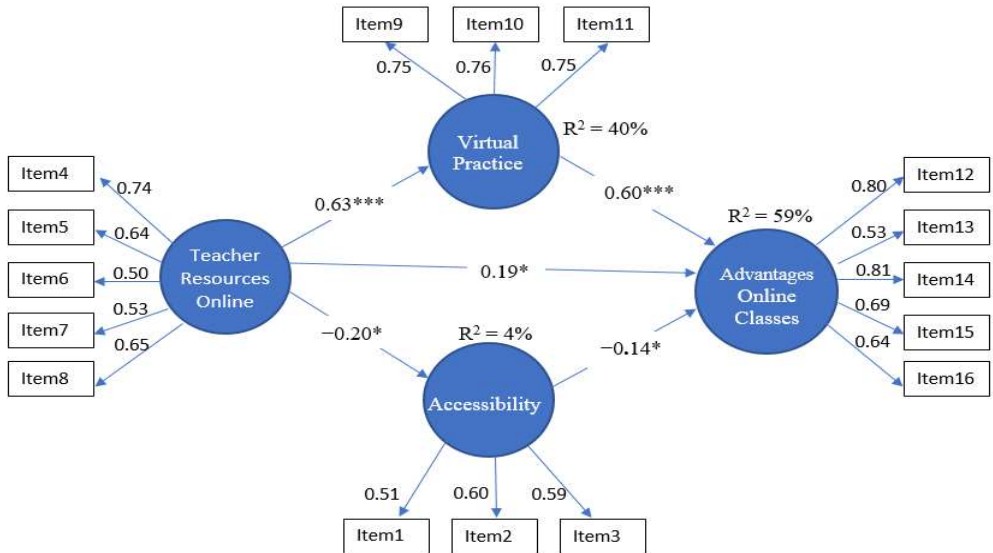

**Figure 3.** Confirmed model of the teacher's use of online resources on advantages of online classes, mediated by virtual practices and by accessibility and resources. * $p < 0.05$, *** $p < 0.001$.

As can be seen in Table 1, the model examined with multiple linear regression analysis is significant ($p < 0.001$) for a statistical power higher than the critical value of 0.80; in addition, the effect size of the regression is significant even for the lower limit because it is greater than 0.25, which is confirmed by Cohen's $f2$ (>0.35). Predictors explain 41.5% of the total variation in the assessment of online classes. On the other hand, the standardized beta weights allow us to observe that virtual practice is the most crucial variable in predicting the valuation of online classes, followed by the teacher's variable use of online resources and third-place accessibility and resources.

**Table 1.** Multiple linear regression model for predicting the assessment of advantages of online classes in biology students.

|  | B (SE) | β | t | p | 95% CI |
|---|---|---|---|---|---|
| (Constant) | 7.083 (0.486) |  | 8.394 | 0.000 | [3.126, 5.039] |
| Use of online resources by the teacher | 0.240 (0.052) | 0.222 | 4.612 | 0.000 | [0.138, 0.343] |
| Accessibility and resources | −0.185 (0.077) | −0.102 | −2.383 | 0.018 | [−0.337, −0.032] |
| Virtual practice | 0.737 (0.074) | 0.481 | 9.959 | 0.000 | [0.591, 0.882] |
| F(3, 328) = 77.472 | $p = 0.000$ | R = 0.644 | $R^2 = 0.415$ [0.334, 0.496] | $f^2 = 0.71$ | $1 - β = 0.99$ |

Table 2 shows that according to the comparison of means with Student's *t*-test, between male and female students, there are no significant differences ($p > 0.05$) or practical differences (d < 0.20) in the assessment of the advantages of online classes.

**Table 2.** Comparison of the evaluation of online classes according to the gender of the student.

|  | n | Mean | SD | t | gl | p | d |
|---|---|---|---|---|---|---|---|
| Female | 184 | 6.39 | 2.934 | 0.848 | 330 | 0.397 | 0.09 |
| Male | 148 | 6.12 | 2.809 |  |  |  |  |

## 4. Discussion

This work aimed to determine the effect of teacher and student variables on the assessment of biological sciences students of a Peruvian public university regarding the advantages of virtual classes. This study, based on self-reports, showed that the valuation of the advantages of virtual classes during the first academic cycle implemented during

the pandemic and the confinement dictated by the Peruvian government was predicted significantly and in a higher proportion by the students' valuation of virtual practices.

A fundamental aspect to understanding this high and significant influence of the perception of virtual practices on the valuation of virtual classes during the pandemic can be centered on the fact that, in biological sciences, classes are essentially theoretical–practical, an educational process in which practices are essential. During the implementation of the virtual modality in scientific disciplines, forced by the social confinement due to COVID-19, even to a lesser extent, educational research has shown the importance and effectiveness of virtual practices in the training of undergraduate and graduate students as well as in the assessment of their effectiveness for learning and academic achievement [34], especially in the biological sciences [33,35–37]. In this sense, our results obtained with structural equation modeling are reaffirmed with the findings of the multiple linear regression analysis, where virtual practice stands out among the predictors as the variable with the greatest explanatory weight for the evaluation of online classes.

Another aspect derived from the results of this study is the significant but negative effect of the factor of accessibility and management of digital resources by the students on the valuation of the advantages of virtual classes. This would mean that, despite the students' difficulties in accessing internet connection and digital resources, students positively value the advantages of virtual classes in the first academic cycle during social confinement due to the COVID-19 pandemic.

Although this result shows a significant but negative predictive relationship, which contrasts with most of the findings regarding the effect of the variables of accessibility and use of digital resources on the assessment and satisfaction concerning virtual classes and their influence on academic achievement, it coincides with the studies that had already shown difficulties in terms of internet accessibility and the use of digital resources, despite the acceptance of virtual classes [14,15,30,41–43].

Similar to our study, perceived limitations of implementing such virtual classrooms have been reported, e.g., lack of strategies and habits in the use of virtual and interactive methodologies [15,41]; lack of network infrastructure, computers, and internet access [30,42,43]; affectation by the network, data card and power outage problems [13]; inadequate e-learning support [14]; and unstable internet connection and university servers used to develop virtual classes [15].

Another aspect to be considered is the differential effect of the teacher's management of digital resources on the factors of Virtual Practices (significant and positive), Accessibility and uses of digital resources by students (significant but negative), and the valuation of the Advantages of virtual classes (significant and positive).

Regarding the effect of the teacher's management of digital resources on virtual practices, our result coincides with the findings in Peruvian graduate students in educational sciences because significant effects of teacher performance concerning teaching and feedback in virtual classes were observed on the participation and relevant practice of students in these interactions [25]. Similarly, in research with Malaysian students, teacher performance had a significant effect on student factors, e.g., the student's evaluative perception of student–teacher interaction and students' awareness of online learning [8].

On the other hand, our results reflected the significant but negative effect of the factor of teachers' management of digital resources and virtual education on the factor referring to the accessibility and use of digital resources by students. This causal type of association may be indicative that a perception of high teacher mastery of digital and virtual education resources predicts difficulties in internet accessibility and low student mastery of digital resources for virtual education [13, 28]. That is, this may be reflecting difficulties that students perceived from virtual classrooms, as reported in other studies on virtual classroom perceptions during the pandemic, which we have mentioned in the previous paragraphs of this section.

Regarding the relationship between the teacher's use of digital resources (teaching competence and performance) and the evaluation of the advantages of virtual classes, our

data proved to be significant, but previous research is not conclusive regarding this relationship. An investigation into the perception of Peruvian graduate students in educational sciences confirmed that the use of digital resources by the teacher had a moderate but significant influence on the evaluation of the advantages of virtual classes [30], but an investigation by Chinese university students reported a significant but negative effect of teacher performance on learning outcomes and satisfaction with online learning [24].

Likewise, these data coincided with findings of the indirect effect of teacher performance in virtual classes on the results of the application of what was learned in virtual classes [25], and the indirect effect of teacher performance in virtual classes on student satisfaction in virtual classes [8].

Even though the objectives formulated were achieved, the study has some limitations such as the use of a non-probabilistic sample of voluntary participants given the context of COVID-19 and the difficulties of access to virtual connectivity for many students; so, it is recommended that generalizations be made with caution. Likewise, the use of self-reports for the measurement of variables could add some social desirability bias in the data, as illustrated by the literature. Despite these limitations, we sought to guarantee the internal validity of the study (primary systematic variance) by reducing any instrumentation problems because, before the analysis of the research results, satisfactory evidence of validity and reliability of the instrument was obtained with rigorous procedures recommended by current measurement theory and the literature specializing in the field of self-report measurement. Likewise, the validity of the statistical conclusions has been guaranteed by complying with type I error analysis ($\alpha$), type II error ($1-\beta$), and effect size. Given the scarce existing scientific knowledge and results that are not consistent with existing findings, the present study broadens and strengthens the notion that the impression generated by the teacher in the efficient management of pedagogical and technological online resources is of great importance to the design and implementation of virtual practices that enable optimal learning, despite the limitations or difficulties of access to digital resources by the students, and as a consequence of all this, the assessment of the advantages of online classes is positive according to the students' perceptions.

## 5. Conclusions

The perception of university students regarding the management of online resources by their professors in a school of biological sciences during the COVID-19 pandemic was a determinant for a favorable assessment of the advantages offered by online classes. This positive impact relationship is conditioned by the positive mediation of Virtual Practices and the negative mediation of Accessibility to online resources. These mediations imply that the greater the opportunities for virtual practices and the lesser the difficulties of accessibility and use of digital resources by students are, the higher the student's valuation of the advantages of virtual classes during the pandemic will be.

In a linear model where the predictor variables was configured additively, virtual practice was the most crucial variable in predicting the assessment of the advantages granted by online classes in biological science subjects. Also, regardless of the gender of the students, the assessment of the advantages of online classes was similar.

**Author Contributions:** Conceptualization, A.B.-R. and H.A.-A.; Methodology, A.B.-R. and W.C.-L.; Validation, W.C.-L.; Formal analysis, A.B.-R.; Investigation, A.B.-R., H.A.-A., R.A.-G. and V.C.-L.; Resources, H.A.-A. and R.A.-G.; Data curation, W.C.-L.; Writing—original draft, A.B.-R. and V.C.-L.; Writing—review & editing, W.C.-L.; Visualization, V.C.-L.; Supervision, A.B.-R. and H.A.-A.; Project administration, R.A.-G. All authors have read and agreed to the published version of the manuscript.

**Funding:** This research received no external funding.

**Institutional Review Board Statement:** The study was conducted in accordance with the Declaration of Helsinki and approved by the Institutional Review Board of Sciences Biological Faculty of Universidad Nacional de San Cristobal de Huamanga.

**Informed Consent Statement:** Informed consent was obtained from all students involved in the study.

**Data Availability Statement:** The data are handled confidentially; however, the interested researcher can send a request to the corresponding author, and the database will be provided.

**Conflicts of Interest:** The authors declare that the research was conducted in the absence of any commercial or financial relationships that could be construed as potential conflicts of interest.

**Appendix A**

QUESTIONNAIRE OF CONDITIONS AND EVALUATION OF ONLINE CLASSES
General Data
Name______________________________ Gender______ Age_______
Specialty ____________________ Year of admission ______
Current Cycle _____________ Date _____________

Dear student, this questionnaire is made to evaluate the quality of online courses. Please, answer honestly. This questionnaire examines the degree of acceptance of students in the Faculty of Biological Sciences of the UNSCH regarding the conditions, difficulties, and advantages of online study in the first semester of 2020-I.

Instructions:

For each of the following statements, mark with the cursor one of the four options that best represents your degree of acceptance or rejection; choose only one of the four options:

0 = Never, 1 = Sometimes, 2 = Almost always, 3 = Always

FACTOR 1: ACCESSIBILITY AND RESOURCES

1 In my home, other people use the network when I take my classes.

2 Due to connectivity problems I have difficulties to receive a good online class.

3 In most of my classes or virtual practices the teachers have connectivity and signal problems.

FACTOR 2: USE OF ONLINE RESOURCES BY THE TEACHER

4 My laboratory practices include various exercises with virtual technological applications.

5 Teachers properly use discussion forums to supplement my learning.

6 When teachers give us tests, they provide timely feedback.

7 In theoretical classes, teachers have a broad command of technological resources.

8 Practice teachers show a broad mastery of technological resources.

FACTOR 3: VIRTUAL PRACTICES

9 Online laboratory practices allow students to apply their knowledge.

10 Virtual practices get students to replicate the demonstration.

11 The virtual modality is effective for learning in laboratory practices.

FACTOR 4: ASSESSMENT OF ONLINE CLASSES

12 The virtual modality is suitable for my learning at the Professional School of Biology.

13 The live virtual contact (synchronous) allows the teacher to explain the class or practice better.

14 Online teaching enables students, a high-quality learning.

15 Virtual assessments allow me to show what I have learned.

16 I have improved my autonomous learning capacity due to the Online modality.

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
