# Peer review of "Perception of Peruvian Students Studying in Biological Sciences about the Advantages of Virtual Classes during the COVID-19 Pandemic"

_education, doi:10.3390/educsci13060626_

Round 1
Reviewer 1 Report
This research is interesting especially for the statistical procedures used to analyse data; however, language should be improved and reviewed by a native speaker.
The English of this article is not adequate; there are also some redundancies along the text.
Author Response
Dear Reviewer 1
We have corrected this problem; a native English academic reviewed and made improvements to our manuscript.
Reviewer 2 Report
The authors carefully studied the experience of similar studies in different countries of the world. The method used, the validity and relevance of the data obtained were very carefully substantiated. In my opinion, the structure of work suffers from the effect of circulation. When reading the work, it becomes difficult to break through the jungle of constant recurrences in the description of the hypothesis, variables to the reasoning associated with the description of these variables in other studies. There is no explanation of what the Structural model of the teacher's use of online resources on advantages (assessment) of online classes, mediated by accessibility and virtual practices in the online study consists of (Figure 3). Therefore, it is not clear how the obtained results are related to the elements of this model. The text mentions the variables of the first line, the second line, latent. But a clear classification with a list of these variables is not presented. Sources are formatted incorrectly 25. Removed for peer review. and 30. Removed for peer review. Is this one and the same?
There is no citation of works from the reference list under numbers 19,20,21,22,41,42,43.
Author Response
… In my opinion, the structure of work suffers from the effect of circulation. When reading the work, it becomes difficult to break through the jungle of constant recurrences in the description of the hypothesis, variables to the reasoning associated with the description of these variables in other studies.
Answer. We have described separately the variables that would be related with the students’ valoration about the advantages of Online classes. Also, We have made some cuts and modifications to the text (in Introduction section).
There is no explanation of what the Structural model of the teacher's use of online resources on advantages (assessment) of online classes, mediated by accessibility and virtual practices in the online study consists of (Figure 3).
Answer. Figure 3 only shows the confirmation of the hypothetical model proposed in Figure 2. From line 292 to line 297 (paragraph restructured in this 2nd version), we describe the hypothetical model put to the test with the modeling four variables were described in Introduction.
While the hypothetical model (Figure 2) clearly shows the type of predictive relationship between the Teacher Resources Online variables perceived by the student body, and Advantage Online Classes perceived by the student body, mediated by Virtual Practices and accessibility perceived by the student body.
Therefore, it is not clear how the obtained results are related to the elements of this model.
Answer. The type of relationships between the variables that we have described in Introduction were taken to a hypothetical model that we have shown in Figure 2. Therefore, the confirmatory results shown in Figure 3 allow us to maintain that said resulting model (Figure 3) corresponds to the results four elements or factors that we have described in Introduction and the type of relationship between these factors (variables) that we have hypothesized in Figure 2.
The text mentions the variables of the first line, the second line, latent. But a clear classification with a list of these variables is not presented.
Answers.
In results (Figure 3) the variables must not be described (or a clear list of these variables should be presented). However, in the Introduction we have elaborated more fully on each of the four factors4e that were included in this study. In its place, in Method we have described the instrument that includes the measurement of these four factors.
Sources are formatted incorrectly 25. Removed for peer review. and 30. Removed for peer review. Is this one and the same? There is no citation of works from the reference list under numbers 19,20,21,22,41,42,43.
Answer. All those references ARE included since the first version of the manuscript, in the Discussion section:
- The quote from references 18 to 22 was modified and appears on line 73.
- The citation to reference 41 is on line 376, and the citation to references 42 and 43 is on line 377, in the Discussion section.
Reviewer 3 Report
Dear author/s,
I was glad to read your paper which develops a contemporary theme and demonstrates interesting findings. The text is well-organized and structured in an appropriate way.
However, I would like to mention some points that should be taken into consideration. To begin with, some corrections regarding the use of the English language and better linking of sentences are needed. The content of the seventh paragraph of the introduction should be also supported by some in-text citations. Finally, some more details regarding ethical considerations and limitations of the research study should be given.
Best regards

Author Response
REVIEWER 3
“I was glad to read your paper which develops a contemporary theme and demonstrates interesting findings. The text is well-organized and structured in an appropriate way. However, I would like to mention some points that should be taken into consideration”.
Answer. We have responded to this observation; the text was reviewed by a native speaker of English.
“To begin with, some corrections regarding the use of the English language and better linking of sentences are needed. The content of the seventh paragraph of the introduction should be also supported by some in-text citations. Finally, some more details regarding ethical considerations and limitations of the research study should be given”.
Answers.
- We have added citations 14-22 to lines 77 and 78, to support what is said in the 7th paragraph of the Introduction.
- We have restructured the ethical consideration part, for this reason, we have placed a special section on Ethical considerations on lines 234 to 241.
- Limitations were included in Discussion. For example, in lines 417 through 427, some limitations are emphasized.
Round 2
Reviewer 1 Report
The article was sufficiently improved.